# T-BACCO SCORE: A predictive scoring tool for tuberculosis (TB) loss to follow-up among TB smokers

**Zatil Zahidah Sharani[1], Nurhuda Ismail [1,2] \*, Siti Munira Yasin[1,2], Muhamad Rodi Isa[1], Asmah Razali[3], Mas Ahmad Sherzkawee[4], Ahmad Izuanuddin Ismail [2]**

**1** Department of Public Health Medicine, Faculty of Medicine, Universiti Teknologi MARA (UiTM), Sungai Buloh Campus, Sungai Buloh, Selangor, Malaysia, **2** Hospital Al-Sultan Abdullah, Universiti Teknologi MARA (UiTM), Bandar Puncak Alam, Selangor, Malaysia, **3** Sector TB/Leprosy, Disease Control Division, Ministry of Health, Putrajaya, Malaysia, **4** Selangor State Health Department, Sector TB/Leprosy, Disease Control Division, Shah Alam, Selangor Darul Ehsan, Malaysia

\* nurhuda169@gmail.com

## Abstract

### Introduction

Loss to follow-up (LTFU) and smoking during TB treatment are major challenges for TB control programs. Smoking increases the severity and prolongs TB treatment duration, which lead to a higher rate of LTFU. We aim to develop a prognostic scoring tool to predict LTFU among TB patients who smoke to improve successful TB treatment outcomes.

### Materials and methods

The development of the prognostic model utilized prospectively collected longitudinal data of adult TB patients who smoked in the state of Selangor between 2013 until 2017, which were obtained from the Malaysian Tuberculosis Information System (MyTB) database. Data were randomly split into development and internal validation cohorts. A simple prognostic score (T-BACCO SCORE) was constructed based on the regression coefficients of predictors in the final logistic model of the development cohort. Estimated missing data was 2.8% from the development cohort and was completely at random. Model discrimination was determined using c-statistics (AUCs), and calibration was based on the Hosmer and Lemeshow goodness of fit test and calibration plot.

### Results

The model highlights several variables with different T-BACCO SCORE values as predictors for LTFU among TB patients who smoke (e.g., age group, ethnicity, locality, nationality, educational level, monthly income level, employment status, TB case category, TB detection methods, X-ray categories, HIV status, and sputum status). The prognostic scores were categorized into three groups that predict the risk for LTFU: low-risk (<15 points), medium-risk (15 to 25 points) and high-risk (> 25 points). The model exhibited fair discrimination with a c-statistic of 0.681 (95% CI 0.627–0.710) and good calibration with a nonsignificant chi-

**Data Availability Statement:** Data used in this study cannot be shared publicly because it is the Malaysian TB database from the national registry.

Data is available upon request from the Communicable Disease Center, Ministry of health Malaysia for researcher who received ethical approval from the Medical Research Ethics Committee. The ethics registered number for this study is NMRR-21-592-58245, further information is available at the official website of the MREC https://www.nih.gov.my/mrec/.

**Funding:** The author(s) received no specific funding for this work.

**Competing interests:** The authors have declared that no competing interests exist.

square Hosmer–Lemeshow's goodness of fit test χ2 = 4.893 and accompanying p value of 0.769.

## Conclusion

Predicting LTFU among TB patients who smoke in the early phase of TB treatment is achievable using this simple T-BACCO SCORE. The applicability of the tool in clinical settings helps health care professionals manage TB smokers based on their risk scores. Further external validation should be carried out prior to use.

## Introduction

Tuberculosis and tobacco smoking are two major public health issues that cause millions of deaths every year. An estimated 830,000 people diagnosed with tuberculosis (TB) were linked to tobacco smoking in 2017 worldwide [1]. A study based on mathematical modeling estimated that smoking could increase the number of TB cases by 18 million between 2010 and 2050 [2]. The effects of smoking are limited not only to lung health but also to the general defensive immune system of humans, which could be why smokers are also at increased risk for extrapulmonary tuberculosis [3]. There is consistent evidence that tobacco smoking is associated with poor TB treatment outcomes and delayed sputum and culture conversion, which indicates a longer period of infectiousness among TB patients who smoke [4, 5] and increases the risk of recurrence after successful anti-TB treatment [6, 7]. Generally, TB cases who were LTFU represent approximately 5.3% of the total TB population, while TB patients who smoke had a higher proportion of loss to follow-up from TB treatment at double to triple rates [8]. This situation warrants significant concern, as these patients are at higher risk for reactivation of TB infections as well as developing multiple drug-resistant tuberculosis (MDR-TB).

TB LTFU is influenced by a myriad of interrelated factors, such as patient beliefs and personal factors, health system and service factors, economic factors, including poverty and gender discrimination, and the social context in terms of social support and TB-related stigma [9–13]. Many of these factors require qualitative input and further exploration of losses to follow-up TB cases, which is beyond the scope of this study. In industrialized urban areas, where smoking and TB are prevalent, the determinants for LTFU consist of low socioeconomic status, working-age population, previously treated TB cases, TB-HIV coinfection, and patients with moderate lesions on chest X-rays [14]. Smokers tend to have a lower socioeconomic status, which suggests potential additional financial barriers for them to travel and complete their TB treatment [15].

The current TB control program in Malaysia utilizes a directly observed treatment (DOT) strategy to manage patients receiving TB treatments and to prevent the LTFU issues among TB patients [16]. Newly diagnosed TB patients must come to the nearest chest clinic to take their TB medication through daily dosing anti-tuberculosis regimens during the intensive phase (1st two months of TB treatment). Continuation of DOT during the maintenance phase depends on the sputum seroconversion status, clinical manifestations, and adherence to TB treatment. Patients with fewer complications during TB treatment and who exhibit good adherence to their medication regimen will be allowed to continue taking their anti-TB drugs at home with supervision by family members, community representatives or NGO volunteers through a modified DOT strategy, while patients who live alone are required to take their medication through self-administered treatments. Then, all patients must attend physical visits for

follow-up at 1- to 2-month intervals at TB clinics [16]. These methods have been practiced for more than a decade, yet the LTFU rate in the country remains high. In addition, patients with a previous history of LTFU will tend to redefault, as the underlying factors leading to this outcome were not identified and remain unsolved [10].

The development of a multivariable prognostic model includes identifying the important predictors, assigning relative weights to each of the predictors and estimating the performance of the model through calibration and discrimination, and its potential for optimism using internal validation techniques [17]. Several predictive models have been developed to identify patients with TB who are at risk for unsuccessful TB treatment outcomes [18]. However, to date, there has been little information or synthesis of the prognostic factors for LTFU among TB patients who smoke. Targeted interventions for high-risk subpopulations is a way to improve the successful treatment outcome rate among the TB population. In this study, we aim to develop a simple prognostic scoring checklist to predict LTFU outcomes among TB patients who smoke using readily available information during initial patient follow-ups at TB clinics. The model can stratify the targeted population into different risk groups for intervention and management prioritization.

## Materials and methods

### Study design

This was a retrospective cohort study that involved the development and internal validation of a prognostic model. We utilized a cohort of confirmed and registered TB patients in Selangor state that was obtained from the National MyTB database version 2.1 from 2013 until 2017. The population of Selangor state was selected because it had recorded the highest number of TB cases in peninsular Malaysia each year and has the highest population percentage in Malaysia (20.1% of the total Malaysian population). Selangor is also one of the urbanized states in peninsular Malaysia, with the highest employment rate and job opportunities, especially in the industrial sector [19].

### Database

The National MyTB database is a national surveillance database that consolidates all data on TB cases in Malaysia and is owned by the Disease Control Division, Ministry of Malaysia. TB is a notifiable disease in Malaysia by law under the Prevention and Control of Infectious Disease Act, 1988 (Act 342) [20]. Every medical practitioner who treats or becomes aware of a TB infection at any location shall notify the case to the nearest health office or government health facility within seven days of diagnosis. TB cases are notified via the tuberculosis information system form (TBIS 10A-1) and recorded in the state MyTB database, which is managed by the state TB task force team. Patients' data in the surveillance database are consolidated at the national level, and TB treatment outcomes were defined and recorded after one year of surveillance. Information available from the database includes patient sociodemographic profiles, past medical and comorbidity profiles, tuberculosis disease profiles, laboratory and radiology information and TB treatment profiles.

### Data processing and eligibility criteria

All adult TB patients aged >18 years who were current smokers at the time of their diagnosis were considered for analysis (Fig 1). For TB cases that were initially registered as TB but were changed to other than TB cases, cases with missing data on treatment outcomes and duplicated cases were removed from the dataset. Cases with multiple drug-resistant TB (MDR-TB) were

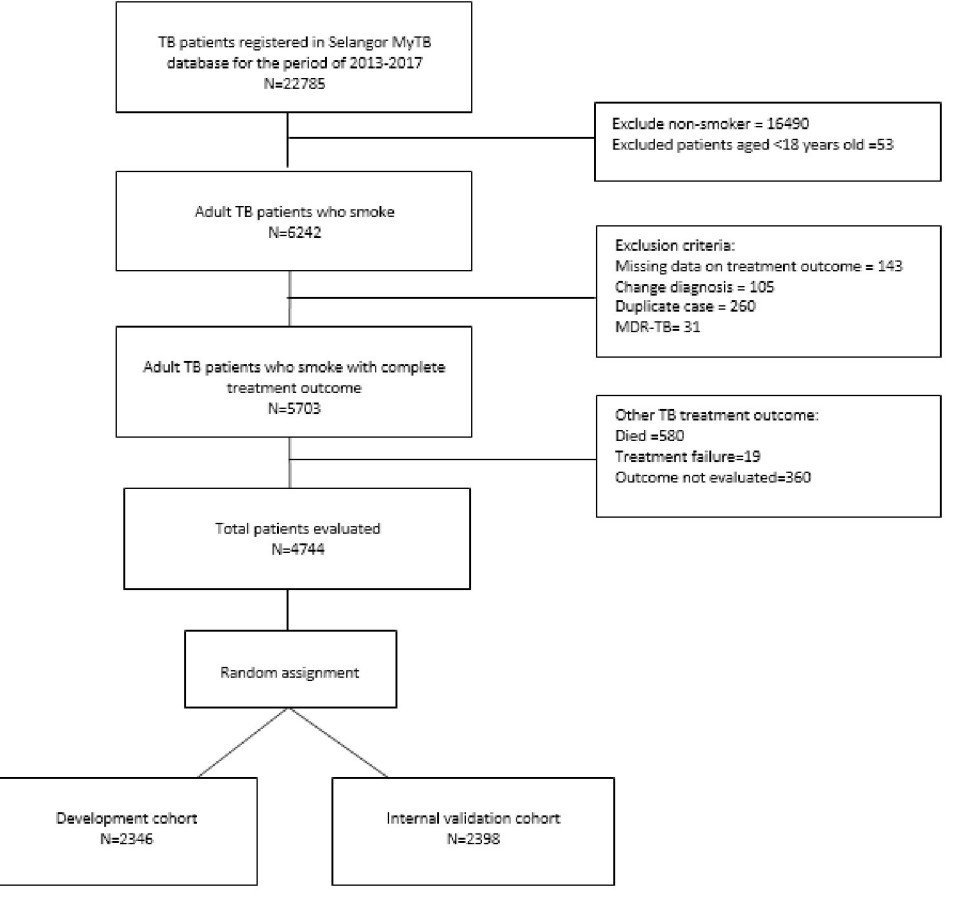

**Fig 1. Data processing flow chart.**

also excluded from the analysis, as the treatment outcome definition for MDR-TB cases is different from the definition for non-MDR-TB. The outcome being measured in this study is the loss to follow-up (LTFU) outcome. Other TB outcomes such as treatment failure, death and outcome not evaluated, were not included in the analysis.

## Operational definition

The classification of TB treatment outcomes are defined according to the Clinical Practice Guideline of Tuberculosis by the Ministry of Health, Malaysia [21] and the definition and reporting framework for TB by the WHO [22] as follows:

1. Loss to follow-up (defaulted in the past): TB patient who did not start treatment or whose treatment was interrupted for two consecutive months or more.

2. Cured: Bacteriologically confirmed TB patient who was subsequently smear-or-culture-negative during the last month of treatment or on at least one previous occasion.

3. Completed treatment: Patient who completed TB treatment without meeting the criteria for cure or treatment failure.

4. Treatment failure: TB patient whose sputum smear or culture was positive at five months or later during TB treatment.

5. Died: TB patient who died for any reason before starting or during TB treatment (all-course mortality)

6. Outcome not evaluated: TB patients with no assigned treatment outcome, including TB cases who were "transferred out" to another country for whom the treatment outcomes were not known.

Successful TB treatment outcomes were cases that were cured and had completed TB treatment, while unsuccessful treatment outcomes included all other outcomes (e.g., loss to follow-up, treatment failure, death and outcome not evaluated). The smoking status in this study was obtained by confirming whether the patient was a "current smoker" or not at diagnosis. A current smoker was defined as a patient who currently smoked at least one tobacco product every day (daily smoker) or less than daily (occasional smoker) [23].

## Statistical analysis

The dataset was randomly divided into two cohorts with a 1:1 ratio using IBM SPSS version 26.0 software. The development cohort was used to develop the prognostic model, while the other cohort in the dataset was used for internal validation. Sociodemographic profiles, disease profiles and patient comorbidity variables were reported as frequencies and percentages. The differences across both cohorts were compared using the chi-square test to ensure homogeneity between the two cohorts. To determine the potential prognostic factors for loss to follow-up, univariate and multiple logistic regression were applied. Variables with P values <0.25 in the simple logistic regression (S-LogR) were fitted into multiple logistic regression models (M-LogR). Weighted points were assigned to each of the final predictors using the linear shrinkage factor formula described by Van Houwelingen and Le Cessie [(model χ2 –df)/ model χ2] [24]. The calculated shrinkage factor using the above formula was multiplied to the beta coefficient value of each factor and were rounded to the nearest integers. The shrunken and rounded method was used to avoid overfitting of the developed model [24, 25].

## Variable selection for the LTFU prognostic model

A total of 2346 patients were included in the univariate analysis. Crude associations of sociodemographic profiles, disease profiles and comorbidities with LTFU were analysed using a simple logistic regression analysis. By using a significance value of p<0.25, the variables that were included in the multiple logistic regression analysis consisted of age, nationality, locality, ethnicity, educational level, monthly individual income, occupation, TB case category, TB anatomical location, TB detection method, chest X-ray status, BCG status, sputum status, and HIV status. For the monthly income level, the cut point at RM2160 (488.30 USD) used was based on the median personal income of population in this country according to the department of statistics Malaysia in 2018 (DOSM). Sex (p = 0.916) and DM comorbidity (p = 0.315) were not included in the multivariable analysis. A backward selection process was conducted to identify the most relevant predictors for outcomes by retaining variables with p values <0.1. Among the 14 variables included in the multiple logistic regression analysis, 12 variables were retained in the final model (e.g., age group, ethnicity, locality, nationality, educational level, monthly income level, working status, TB case category, TB detection methods, chest X-ray categories, HIV status and sputum status).

The discrimination of the predictive model was evaluated by using the area under the receiver operating characteristic (ROC) curve (AUC). AUC value ranges from 0.5 to 1, where a greater AUC value indicates that the model has better ability to distinguish patients who were LTFU and had successful TB treatment outcomes [26]. Model calibration (predictive

accuracy) was determined by the nonsignificant Hosmer–Lemeshow goodness of fit test and visualized in the calibration plot [27]. The performance of the development model was validated based on internal validation cohort dataset. To facilitate the applicability of the model in clinical settings, a risk-scoring table was constructed. Risk scores were calculated for each patient in the development cohort according to the weighted points for each predictor. The summed score for each patient was categorized into three groups based on percentiles of 33.33%, 66.66% and 100.00%, which were significantly distinct in predicting risks for LTFU. This study was conducted in accordance with the transparent reporting of a multivariable prediction model for individual prognosis or diagnosis (TRIPOD) checklist as a guide [28].

## Ethical considerations

This was a retrospective study that involved secondary data analysis from the National MyTB database version 2.1 from 2013 to 2017 by the Disease Control Division, Ministry of Health. Thus, no consent from respondents was needed. A formal request for data utilization was provided to the Disease Control Division, Ministry of Health Malaysia prior to study initiation. Registered TB cases were kept anonymous and given a unique identification number. This research involves no more than minimal risk to the subjects; thus, the requirement to obtain informed consent to participate was waived by the Research Ethics Committee (REC/04/ 2020), Universiti Teknologi MARA (UiTM) and Malaysia Ministry of Health Medical Research Ethics Committee (reference number NMRR-21-592-58245).

## Results

A total of 22785 TB patients were registered in the Selangor MyTB database from 2013 until 2017. A proportion of adult TB patients who smoked, 27.4% (N = 6242), was extracted from the database, and 14.0% of the patients who met the exclusion criteria were excluded from the dataset. Cases were randomly split into two cohorts with a 1:1 ratio, where 2346 cases were included in the development cohort and 2398 cases were included in the internal validation cohort. The data processing flowchart is shown in Fig 1. The comparative characteristics of patients in the development and internal validation cohorts are summarized in Table 1. Both cohorts were homogeneous with nonsignificant differences in all variables with p values >0.05 using the chi-square test.

### General characteristics of adult TB patients in the development cohort who smoked

The majority of included patients were aged <50 years old (70.4%), male (95.6%), Malaysian citizens (89.2%), lived in urban areas (83.4%) and were of Malay ethnicity (57.2%). They were mostly employed (63.36%), had monthly incomes of less than RM2160 (87.9%) and attained education to at least the secondary school level (63.8%). In terms of disease profiles, more than 80% of the patients were new cases, had pulmonary TB and were detected through passive case detection. Their chest X-ray status mostly indicated no lesions to minimal lesions (68.5%), and they had positive sputum smears (69.2%). Only 20% of TB patients who smoked had diabetes mellitus (DM), and fewer than 10% tested positive for HIV. Regarding treatment management, 93.2% of TB patients in the development cohort who smoked were on directly observed therapy (DOT) during the intensive phase; however, the percentage dropped to 79.5% during the continuation phase.

A descriptive analysis of the treatment outcomes among the overall adult TB patients who smoked (N = 5703) showed that 69.1% had successful treatment outcomes (e.g., cured = 2517

**Table 1. Comparison of the characteristics of adult TB patients in the development and validation cohorts who smoked.**

| Sociodemographic characteristic | | Development cohort (N = 2346) | | Internal validation cohort (N = 2398) | | |
|---|---|---|---|---|---|---|
| | | N | % | N | % | P value* |
| Age | <50 | 1652 | 70.4 | 1674 | 69.8 | 0.647 |
| | ≥50 | 694 | 29.6 | 724 | 30.2 | |
| Sex | Male | 2243 | 95.6 | 2301 | 96.0 | 0.554 |
| | Female | 103 | 4.4 | 97 | 4.0 | |
| Nationality | Malaysian | 2093 | 89.2 | 2149 | 89.2 | 0.654 |
| | Non-Malaysian | 253 | 10.8 | 249 | 10.8 | |
| Locality | Urban | 1956 | 83.4 | 1982 | 82.7 | 0.507 |
| | Rural | 390 | 16.6 | 416 | 17.3 | |
| Ethnicity | Malay | 1343 | 57.2 | 1383 | 57.7 | 0.589 |
| | Chinese | 354 | 15.1 | 385 | 16.1 | |
| | Indian | 308 | 13.1 | 289 | 12.1 | |
| | Others | 341 | 14.5 | 341 | 14.2 | |
| Education level | Higher education | 396 | 16.9 | 388 | 16.2 | 0.776 |
| | Secondary school | 1496 | 63.8 | 1564 | 65.2 | |
| | Primary school | 201 | 8.6 | 196 | 8.2 | |
| | No education | 253 | 10.8 | 250 | 10.4 | |
| Personal income | ≤RM2160 | 2062 | 87.9 | 2078 | 86.7 | 0.201 |
| | >RM2160 | 284 | 12.1 | 320 | 13.3 | |
| Working status | Working | 1486 | 63.3 | 1529 | 63.8 | 0.764 |
| | Not working | 860 | 36.7 | 869 | 36.2 | |
| **Disease profile** | | | | | | |
| TB anatomical location | Pulmonary TB | 2080 | 88.7 | 2140 | 89.2 | 0.524 |
| | Extra-Pulmonary TB | 266 | 11.3 | 258 | 10.8 | |
| TB case category | New case | 2158 | 92.0 | 2201 | 91.8 | 0.799 |
| | Recurrent case | 188 | 8.0 | 197 | 8.2 | |
| TB case detection | Active | 80 | 3.4 | 85 | 3.5 | 0.800 |
| | Passive | 2266 | 96.6 | 2313 | 96.5 | |
| BCG scar | Yes | 2050 | 87.4 | 2097 | 87.4 | 0.946 |
| | No | 296 | 12.6 | 301 | 12.6 | |
| X-ray status (N = 4678) | No lesion to minimal lesion | 1586 | 68.5 | 1613 | 68.3 | 0.889 |
| | Moderate to severe lesion | 730 | 31.5 | 749 | 31.7 | |
| Sputum status (N = 4668) | Negative | 710 | 30.8 | 720 | 30.5 | 0.820 |
| | Positive | 1596 | 69.2 | 1642 | 69.5 | |
| HIV status | Positive | 156 | 6.6 | 173 | 7.2 | 0.197 |
| | Negative | 2119 | 90.3 | 2171 | 90.5 | |
| | Test not done | 71 | 3.0 | 54 | 2.3 | |
| DM status | Yes | 470 | 20 | 500 | 20.9 | 0.486 |
| | No | 1876 | 80 | 1898 | 79.1 | |
| DOT during the intensive phase | Yes | 2187 | 93.2 | 2244 | 93.6 | 0.622 |
| | No | 159 | 6.8 | 154 | 6.4 | |
| DOT during the continuation phase | Yes | 1866 | 79.5 | 1982 | 80.4 | 0.574 |
| | No | 480 | 20.5 | 470 | 19.6 | |

*Test used: Comparison of the characteristics between the development and validation cohorts were conducted using the chi-square test.

or completed treatment = 1424) and 30.9% had unsuccessful treatment outcomes (e.g., loss to follow-up = 803, died = 580, treatment failure = 19 or outcome not evaluated = 360).

## Performance of the final model

A total of 2280 of 2346 cases were analyzed in the logistic regression model with 2.8% (n = 66) of missing information in the development cohort. Out of 16 variables analysed in the univariate analysis (Table 2), 14 variables with p-value >0.25 were included in the multiple logistic regression and 12 variables were retained in the final model (Table 3). The final logistic model had fair discrimination with a c-statistic of 0.681 (95% CI 0.657–0.713). There was a nonsignificant chi-square Hosmer–Lemeshow goodness of fit test value, $\chi 2 = 4.893$, and the accompanying p value was 0.769, which indicated good calibration. These results are comparable to the internal validation model with a c-statistic of 0.668 (95% CI 0.639–0.698) and nonsignificant Hosmer–Lemeshow goodness of fit test with $\chi 2 = 2.223$ and (p = 0.563) The AUCs between the two models (development and the internal validation cohort) showed an overlapping confidence interval indicates that the difference between the curve areas is not statistically significant (Fig 2). A graphical assessment of calibration (calibration curve) of the final model and the internal validation are shown in Fig 3A and 3B.

## Prognostic scoring system for LTFU

Twelve variables were used to develop the LTFU score among TB patients who smoked (T-BACCO SCORE). Weighted points were assigned to each of the prognostic factors as shown in Table 4. The total point scores are normally distributed and range from 4 to 59 (mean = 26.04, SD = 7.10). They were divided into three categories based on percentiles (e.g., 33.33rd, 66.66th and 100th). The low-risk group has scores <15 points, the medium-risk group has scores from 15 to 25 points and the high-risk group has scores > 25 points, as summarized in Table 5. Most patients who were lost to follow-up in the development cohort were in the high-risk group (74.7%). Compared with low-risk patients, patients in the medium and high-risk groups had greater odds for loss to follow-up during TB treatment with OR 2.533 and OR 6.717, respectively. In the development cohort, each risk category was significantly different in predicting the risk for loss follow-up with $\chi 2(2) = 81.94$ and p value of <0.001.

The percentage of missing data for all 16 variables was 2.8% in the development cohort and 2.9% in the internal validation cohort. Variable with incomplete data were the sputum status and chest x-ray status (both are categorical data). The little's MCAR test showed a non-significant p-value in both development cohort (p = 0.516) and internal validation cohort (p = 0.510) indicating that the missing data is MCAR (missing completely at random) and no patterns exist in the missing data [29]. No data imputation was done for cases with missing data in both the development and validation cohort since the missing data number is insignificant (less than 5%) [30, 31]. Model development and internal validation were performed using a complete case analysis.

Additional analyses were conducted by including the directly observed therapy (DOT) variable during the intensive phase and DOT during the continuation phase in the model. By including the DOT variable during the intensive phase in the model, the discrimination value improved to 0.763 (95% CI 0.745–0.780) however the calibration value of the model was unacceptable with significant Hosmer–Lemeshow goodness of fit test. Another analysis trial was conducted by including both variables of DOT during the intensive phase and DOT during the continuation phase in the model, and this model showed an excellent c-statistics value with an AUC value of 0.936 (95%CI 0.925–0.946) and good calibration effect with a nonsignificant Hosmer–Lemeshow goodness of fit test. A model with p values >0.05 for calibration

**Table 2. Crude associations among different potential factors in the development cohort (N = 2346).**

| Variables | | Loss to follow-up | Successful outcome | Crude OR | 95% CI | | P Value* |
|---|---|---|---|---|---|---|---|
| | | N (%) | N (%) | | Lower | Upper | |
| Age (Years) | <50 | 315 (78.6) | 1337 (68.7) | 1.666 | 1.288 | 2.154 | <0.001 |
| | ≥50 | 86 (21.4) | 608 (31.3) | ref | | | |
| Sex | Male | 383 (95.5) | 1860 (95.6) | 0.972 | 0.578 | 1.636 | 0.916 |
| | Female | 18 (4.5) | 85 (4.4) | ref | | | |
| Nationality | Malaysian | 370 (92.3) | 1723 (88.6) | 1.538 | 1.039 | 2.276 | 0.032 |
| | Non-Malaysian | 31 (7.7) | 222 (11.4) | ref | | | |
| Locality | Urban | 346 (86.3) | 1610 (82.8) | 1.309 | 0.962 | 1.781 | 0.087 |
| | Rural | 55 (13.7) | 335 (17.2) | ref | | | |
| Ethnicity | Malay | 239 (59.5) | 1104 (56.8) | ref | | | |
| | Chinese | 36 (9.0) | 318 (16.3) | 0.523 | 0.361 | 0.759 | 0.001 |
| | Indian | 77 (19.2) | 231 (11.9) | 1.540 | 1.148 | 2.065 | 0.004 |
| | Others | 18 (4.5) | 292 (15.0) | 0.775 | 0.555 | 1.082 | 0.134 |
| Education Level | Higher education | 54 (13.5) | 357 (18.4) | ref | | | |
| | Secondary school | 34 (8.5) | 1222 (62.8) | 2.052 | 1.438 | 2.929 | <0.001 |
| | Primary school | 274 (68.3) | 167 (8.6) | 1.864 | 1.136 | 3.058 | 0.014 |
| | No formal education | 39 (9.7( | 199 (10.2) | 2.482 | 1.589 | 3.883 | <0.001 |
| Individual Income (Ringgit Malaysia) | ≤ RM2160 | 382(95.3) | 1680 (86.4) | 3.171 | 1.965 | 5.118 | 0.001 |
| | > RM2160 | 19 (4.7) | 265 (13.6) | ref | | | |
| Working status | Not Working | 172 (42.9) | 688 (35.4) | 1.372 | 1.103 | 1.707 | 0.005 |
| | Working | 229 (57.1) | 1257 (64.6) | ref | | | |
| Site of TB | Extrapulmonary | 38 (9.5) | 228 (11.7) | 1.268 | 0.884 | 1.821 | 0.197 |
| | Pulmonary | 363 (90.5) | 1717 (88.3) | ref | | | |
| TB categories | New case | 339 (84.5) | 1819 (93.5) | ref | | | |
| | Previously treated cases | 62 (15.5) | 126 (6.5) | 2.640 | 1.907 | 3.656 | <0.001 |
| TB detection | Passive | 378 (94.3) | 1888 (97.1) | ref | | | |
| | Active | 23 (5.7) | 57 (2.9) | 2.015 | 1.227 | 3.312 | 0.006 |
| BCG status | Yes | 358 (89.3) | 1692 (87.0) | 1 | | | |
| | No | 43 (10.7) | 253 (13.0) | 0.803 | 0.570 | 1.132 | 0.210 |
| Chest X-ray status (n = 2320) | No to Minimal lesion | 250 (63.1) | 1336 (69.6) | 1 | | | |
| | Moderate to Severe lesion | 146 (31.7) | 584 (30.4) | 1.336 | 1.066 | 1.675 | 0.012 |
| Sputum status (N = 2306) | Positive | 291 (74.2) | 1305 (68.2) | 1.345 | 1.051 | 1.719 | 0.018 |
| | Negative | 101 (25.8) | 609 (31.8) | 1 | | | |
| HIV status | Negative | 349 (87.0) | 1770 (91.0) | 1 | | | |
| | Positive | 44 (11.0) | 112 (5.8) | 1.992 | 1.38 | 2.876 | <0.001 |
| | Test not done | 8 (2.0) | 63 (3.2) | 0.644 | 0.306 | 1.356 | 0.247 |
| DM status | No | 328 (81.8) | 1548 (79.6) | 1 | | | |
| | Yes | 73 (18.2) | 397 (20.4) | 0.868 | 0.658 | 1.144 | 0.315 |

* Test used: Simple logistic regression analysis. P value <0.25 from the univariate analysis were considered in the final logistic model.

(Hosmer–Lemeshow test) and relatively higher value of the c-statistic (area under the ROC curve) indicated relatively better-performing models. The ROC comparison between these models is shown in Fig 4. This finding emphasizes the importance of DOT in both the intensive and maintenance phases for all TB patients who are receiving treatment. Looking back at the objective of this study is to develop a simple prognostic scoring tool for TB patients who smoke to predict their risk for loss to follow-up at the initial phase of TB treatment, preferably

**Table 3. Prognostic scores for loss to follow-up among adult TB patients who smoke according to a multivariate analysis of the development cohort.**

| Variable | | B-coefficient | AOR | 95% CI | p value* |
|---|---|---|---|---|---|
| Age (Years) | <50 | 0.509 | 1.683 | 1.268–2.234 | <0.001 |
| | ≥50 | ref | | | |
| Ethnicity | Malay | ref | | | |
| | Chinese | -0.736 | 0.481 | 0.322–0.718 | 0.001 |
| | Indian | 0.246 | 1.286 | 0.939–1.760 | 0.117 |
| | Others | 0.034 | 0.959 | 0.545–1.688 | 0.885 |
| Nationality | Malaysian | 0.819 | 2.268 | 1.152–5.831 | 0.021 |
| | Non-Malaysian | ref | | | |
| Locality | Urban | 0.473 | 1.604 | 1.163–2.302 | 0.005 |
| | Rural | ref | | | |
| Education Level | No formal education | 1.048 | 2.851 | 1.711–5.092 | <0.001 |
| | Primary school | 0.692 | 1.998 | 1.166–3.611 | 0.013 |
| | Secondary school | 0.569 | 1.797 | 1.202–2.686 | 0.004 |
| | Higher education | ref | | | |
| Individual Income (Ringgit Malaysia) | ≤ RM2160 | 0.764 | 2.146 | 1.293–3.563 | 0.003 |
| | > RM2160 | ref | | | |
| Working status | Not Working | 0.232 | 1.261 | 1.036–1.721 | 0.043 |
| | Working | ref | | | |
| TB case categories | New case | ref | | | |
| | Previously treated cases | 0.672 | 1.956 | 1.380–2.779 | <0.001 |
| TB detection | Passive | ref | | | |
| | Active | 0.530 | 1.843 | 1.024–3.316 | 0.041 |
| X-ray status | No to minimal lesions | ref | | | |
| | Moderate to severe lesions | 0.221 | 1.372 | 1.070–1.758 | 0.013 |
| HIV status | Positive | 0.497 | 1.643 | 1.091–2.475 | 0.017 |
| | Test not done | -0.099 | 0.906 | 0.415–1.979 | 0.804 |
| | Negative | ref | | | |
| Sputum status | Positive | 0.266 | 1.304 | 1.006–1.692 | 0.045 |
| | Negative | ref | | | |

*Test used: Multiple Logistic Regression Analysis (Method Backward LR; B Constant = -4.779, Model assumptions are met, interaction considered in the model and no multicollinearity

Note: AOR = Adjusted odds ratio, CI = Confidence interval, B = Regression coefficient,

Final model performance: c-statistics, AUC = 0.681 (95%CI: 0.652–0.710), Nagelkerke $R^2$ = 0.097, Hosmer and Lemeshow goodness of fit test $\chi2$ = 4.893 and p = 0.769.

after being diagnosed with TB. However, the DOT variables can only be determined once patients have completed their treatments in both the intensive and maintenance phases. Therefore, the variable selection was mainly based on information that was readily available during initial patient consultation sessions with their health care practitioner, which justifies the selection of our final model.

## Discussion

We developed a prognostic model to predict LTFU from treatments among TB patients who smoked in a region where TB infections are endemic. The scores allocated to each predictor allow health care personnel to stratify patient risks for LTFU at the time of diagnosis, as this model utilizes easily accessible information on patient profiles and their disease profiles during

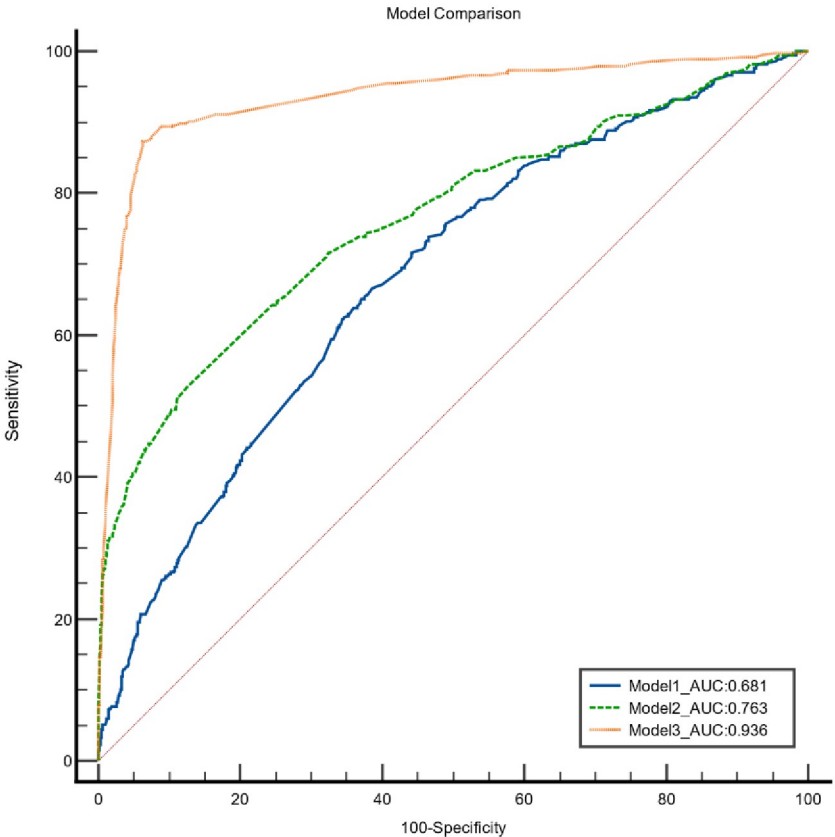

**Fig 2. Comparison of the ROC curve between development cohort and internal validation cohort.** The blue line represents the ROC curve of the development cohort, and the dotted green line is the ROC curve of the internal validation cohort. AUC for development cohort: 0.681 (95% CI 0.657–0.713). AUC for internal validation cohort: 0.668 (95% CI 0.639–0.698).

initial routine follow-up visits at the TB clinic. The variables included consist of age, ethnicity, locality, nationality, educational level, monthly income level, employment status, TB case category, TB detection methods, chest X-Ray categories and HIV status. The final model provides good discrimination results and exhibits good calibration effect with insignificant variations in the internal validation. The results obtained from the final statistical model suggest that this prognostic model is valid for evaluating the probability of LTFU among TB patients who smoke in the early phase of TB treatment initiation. The factors that contributed to high scores (>5 points) for the risk of LTFU in this study included no formal education or primary school education levels, patients living in urban areas, patients with low monthly income levels and previously treated cases. This indicates that low socioeconomic status, urban living conditions and past TB infections play important roles in predicting loss to follow-up in current TB treatments among smokers.

The independent prognostic factors contained in the final model used in this study were consistent with evidence from a previous study regarding the factors associated with LTFU among TB patients [9–13, 32]. Several predictive models for TB treatment LTFU are available from other countries; for example, in Spain, those who live alone or in an institution, are immigrants, intravenous drug user (IVDU), patients with poor knowledge of TB and previously treated TB cases were identified as predictors for LTFU [33]. Another study from Brazil

(a)

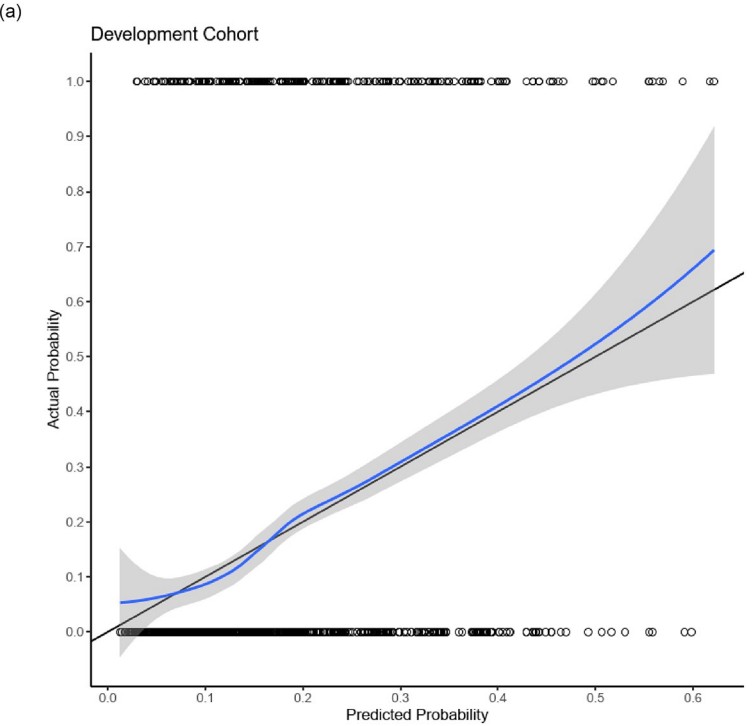

(b)

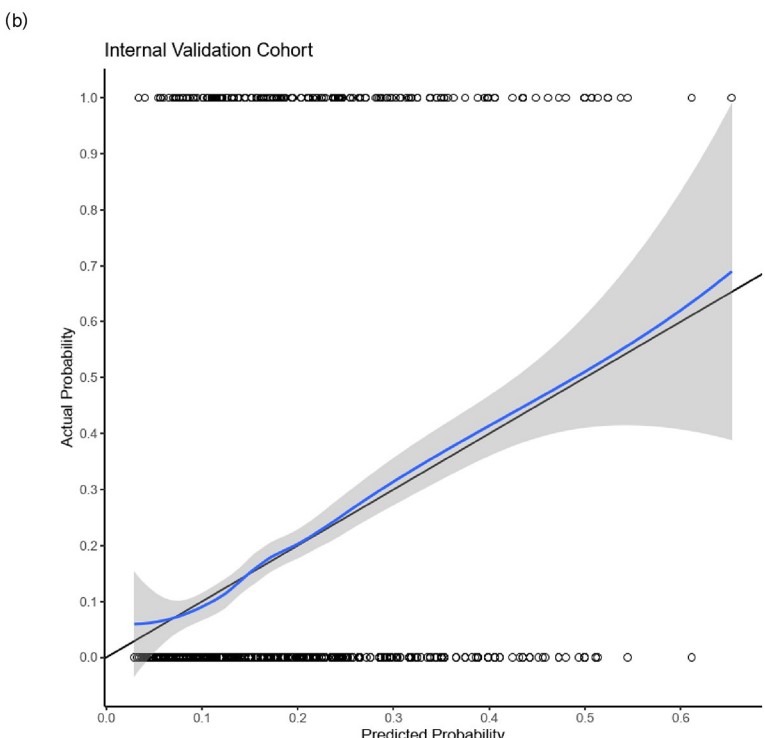

**Fig 3. Calibration curve of the final and internal validation model.** The straight line represents a perfect prediction by an ideal model, and the blue line represents the performance in the final model (A) and internal validation model (B). This visual curve suggests a good calibration between the predicted and the actual outcome in both models.

**Table 4. Rounded scores for each prognostic factor.**

| Prognostic factor | | B-coefficient | Score (B-coefficient*0.88) | Rounded Score |
|---|---|---|---|---|
| Age (Years) | <50 | 0.509 | 0.394 | 4 |
| | ≥50 | ref | 0.000 | 0 |
| Ethnicity | Malay | ref | 0.000 | 0 |
| | Chinese | -0.736 | -0.624 | -6 |
| | Indian | 0.246 | 0.246 | 2 |
| | Others | 0.034 | 0.002 | 0 |
| Locality | Urban | 0.819 | 0.721 | 7 |
| | Rural | ref | 0.000 | 0 |
| Nationality | Malaysian | 0.473 | 0.434 | 4 |
| | Non-Malaysian | ref | 0.000 | 0 |
| Education Level | No formal education | 1.048 | 0.952 | 10 |
| | Primary school | 0.692 | 0.633 | 6 |
| | Secondary school | 0.569 | 0.516 | 5 |
| | Higher education | ref | 0.000 | 0 |
| Individual Income (Ringgit Malaysia) | ≤ RM 2160 | 0.764 | 0.645 | 6 |
| | >RM 2160 | ref | 0.000 | 0 |
| Working status | Not Working | 0.232 | 0.254 | 3 |
| | Working | ref | 0.000 | 0 |
| TB categories | New case | ref | 0.000 | 0 |
| | Previously treated cases | 0.672 | 0.597 | 6 |
| TB detection | Passive | ref | 0.000 | 0 |
| | Active | 0.530 | 0.538 | 5 |
| X-ray status | No to minimal lesions | ref | 0.000 | 0 |
| | Moderate to severe lesions | 0.221 | 0.278 | 3 |
| HIV status | HIV -ve | ref | 0.000 | 0 |
| | HIV +ve | 0.497 | 0.437 | 4 |
| | Test not done | -0.099 | -0.087 | 0 |
| Sputum status | Positive | 0.266 | 0.234 | 2 |
| | Negative | ref | 0.000 | 0 |

Prognostic score = 4*[Age<50] - 6*[Chinese] + 2*[Indian] + 7*[Urban] + 4*[Malaysian] + 10*[No formal education + 6*[Secondary school] + 5*[Higher education] + 6*[Income ≤2160] + 3*[Not working] + 6*[Previously treated case] + 5*[Active case detection] + 3*Moderate to severe X-ray lesion] + 4*[HIV-positive] + 2*[Sputum +ve]

**Table 5. Loss to follow-up by risk group among TB patients who smoked and model performance for the development cohort and internal validation cohort.**

| Risk group | Development cohort (N = 2346) | | | Internal Validation cohort (N = 2398) | | |
|---|---|---|---|---|---|---|
| | N (%) | Loss to follow-up (%) | OR*(95% CI) | N (%) | Loss to follow-up (%) | OR* (95% CI) |
| Low risk (<15 SCORE) | 94 (4.1) | 4 (1.0) | 1 | 109 (4.7) | 8 (2.1) | 1 |
| Medium risk (15–25 SCORE) | 929 (40.7) | 94 (24.3) | 1.526 (0.689–3.380) | 922 (39.6) | 98 (25.2) | 1.502 (0.709–3.178) |
| High risk (>25 SCORE) | 1257 (55.1) | 289 (74.7) | 4.358 (1.995–9.520) | 1047 (47.8) | 281 (72.6) | 3.488 (1.678–7.252) |
| Incomplete data | 66 (2.8%) | | | 69 (2.9%) | | |
| **Discrimination assessment** | | | | | | |
| AUC | 0.681 (0.652–0.710) | | | 0.668 (0.639–0.698) | | |
| **Calibration assessment** | | | | | | |
| Hosmer–Lemeshow's goodness of fit test | 0.769 | | | 0.973 | | |

*Test used: Comparisons of the risk to loss to follow-up between risk groups were conducted using a simple logistic regression test.

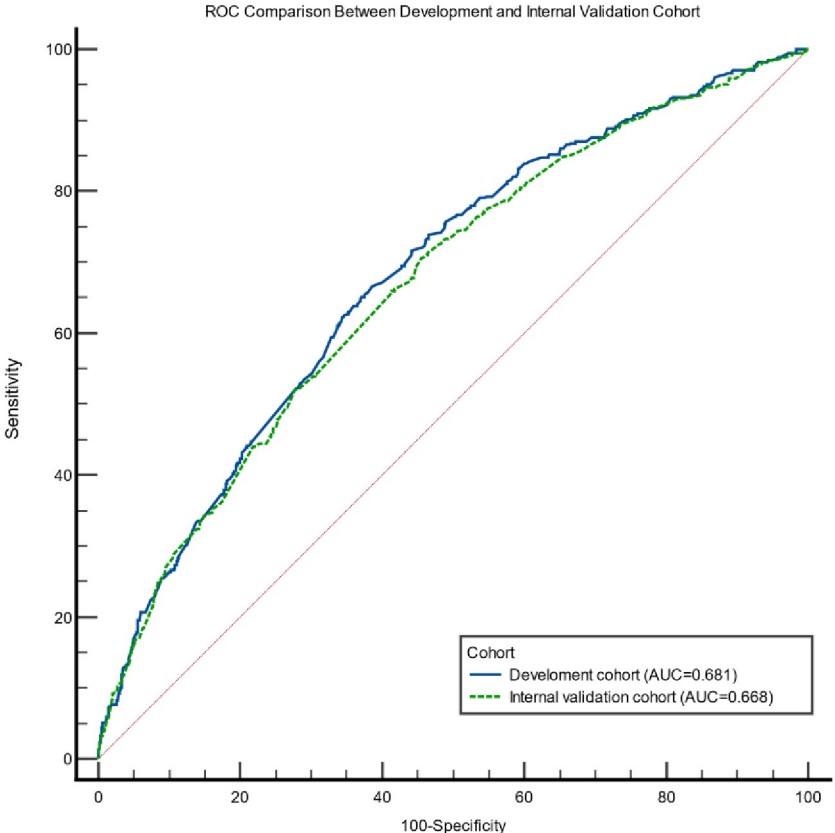

**Fig 4. Comparison of the ROC curve between different model in the development phase.** The blue line represents the ROC curve of the final model (Model1), the dotted green line represents the model that includes variable 'DOTS during intensive phase' (Model2), and the dotted orange line represent the model that include variables 'DOTS during the intensive phase' and 'DOTS during the maintenance phase' (Model3). The AUC value for the model were 0.681, 0.763 and 0.936 respectively. ROC, receiver operating characteristic; AUC, area under the curve; DOTS, directly observe therapy short course.

that used the Classification and Regression Trees (CART) prediction model ranked several important variables that contribute to LTFU, such as the number of doses taken during treatment, age, total number of people in the household, non-IVDU and HIV-negative patients [34]. In comparison with other prognostic models for loss to follow-up among general TB patients [33, 34], our model identifies several predictive factors that are significant for TB smokers, such as young adults (age <50 years old) and HIV-positive individuals. Young adult who are in their working age population are mostly associated with smoking behavior and are at risk for LTFU from TB treatment [8, 14, 35]. While, smoking among TB-HIV patients had a synergistic impact on the burden of both TB and HIV, as smoking leads to worse TB treatment outcomes and inhibits the effectiveness of life-saving ART(anti-retroviral treatment), which worsens the overall health outcomes of patients [36]. At the same time, financial limitations and poor social support aggravated the issues of treatment interruption and LTFU among TB-smoking patients with HIV [37]. It also reflects the performance of our management and control activities at the ground level for TB-HIV patients under the National TB and HIV/AIDS control program [14].

Noncompliance with anti-tuberculosis treatments and LTFU among TB patients are serious issues in our TB control program, yet this is preventable. In Selangor state itself, the rate of

LTFU among general TB patients ranged from 5.8% to 13% from 2014 to 2017 [14]. This rate is higher than the national target rate for loss to follow-up of <2% (NSPTB 2015–2020) [38]. In addition to directly observed therapy (DOT) or directly observed therapy short course (DOTS) as the standard patient management approaches for TB, several supporting actions have been undertaken to decrease the rate of LTFU among TB patients, including video observed therapy (VOT) and financial incentives [39]. VOT has been implemented in some health care facilities since 2019 in Malaysia not only for TB patients but also for managing other chronic diseases to improve patient adherence to treatment. We have been focusing on methods to ensure treatment completion, but we have not reached people who are vulnerable and patients who are at risk of TB and developing poor outcomes. Risk identification and risk management would be ways to supplement the current TB patient management methods to ensure that all patients complete their treatment successfully [40].

At present, no specific management procedure is described for TB patients who smoke in the Clinical Practice Guideline (CPG) [39]. An integrated TB and tobacco control program should be implemented through the health care system to improve overall TB treatment outcomes. Studies have shown that smoking cessation intervention using behavioural therapies (i.e., brief advice from a physician or individual or group counselling sessions) is effective in reducing the loss to follow-up rate, improving recovery, and preventing treatment failure [41–43]. A local study in Malaysia, as part of the SCIDOT project by Awaisu et al. [44], found that integrated TB tobacco treatment using DOTS plus smoking cessation intervention could also improve overall quality of life outcomes (HRQoL) among TB patients who were smokers during the 6-month treatment duration compared to those who received conventional TB care. It is understood that implementing a new program or additional actions in TB management require additional resources such as workforce levels, money, training, and promotional activities. However, if we are committed to achieving the goal of "zero TB cases by 2035", we must be willing to invest more. Human resources are a crucial determinant in managing TB, especially in high-burden states such as Selangor [45]. Human resources not only refer to doctors but also include other supporting personnel such as nurses, medical assistants and other positions that help in managing TB cases. All working staff require proper training to provide quality service to all TB patients, which means that larger budget allocations need to be reserved for training and educational purposes.

In line with the recommendation from the Global Plan to End TB 2018–2022, we are committed to accelerate the innovation of new tools to be used in care delivery to combat TB [46]. This screening tool helps to stratify and prioritize high-risk TB patients who smoke and require specific intervention. Other than that, the score estimated in our study is calculated based on patient's characteristics and disease profiles of a large cohort of TB patients under the current TB controlled programme and healthcare system, which may indicate where improvement can be made. At the point when a person is diagnosed with tuberculosis would be a starting point or "golden opportunity" to advise TB patients to quit smoking, and DOT is an effective way to encourage and support the pathway to quitting smoking. Most smokers with TB are receptive to quitting advice; however, they often begin smoking again when they feel better, which justifies the requirement of proper smoking cessation interventions along with their DOT [47, 48]. This is also part of our initiative to reduce the financial catastrophes faced by families and patients that can be avoided by quitting their smoking behaviour.

A limitation in our study is that the model was developed using secondary data obtained from a surveillance database. Thus, the variables included in the model were limited to the patient information that was available in this database. Other important predictors, such as social support (e.g., living conditions, numbers of people in households and marital status), knowledge of TB infections and TB treatment details (e.g., drug dosage, duration, and side

effects from the drugs), were unavailable. This would create further research opportunities for an updated LTFU prognostic model in the future [49]. However, reflecting on our initial objective in this study, it is unfeasible to obtain all the above information to assess patients on their risk for LTFU at the initial phase of the treatment. For example, the treatment durations and side effects from TB drugs can only be assessed after a certain treatment period, while patient knowledge of TB would require a separate assessment session.

Improving treatment compliance and preventing LTFU among TB patients who smoke is a great challenge that should be addressed through strong support and engagement from political will, health care providers, family, and social organizations. Smoking cessation interventions using the quit smoking program available in the country must be integrated into the TB care routine through appropriate channels. Our early screening tool to predict LTFU among TB patients who smoke will guide health care personnel to identify which smoking TB patients require further intervention during the 6 months of TB treatment, and key activities under the National TB control program must act on the identified prognostic factors highlighted from this study.

## Conclusions

This simple prognostic score (T-BACCO SCORE) facilitates risk stratification for TB patients who smoke for LTFU from TB treatment and allows personalized TB monitoring and management options. Further external validation using the tool across the population shall be conducted prior to its application in clinical settings.

## Acknowledgments

The authors would like to thank the Director General of Health, Malaysia for permission to publish this paper. We would also like to express our gratitude to all health care personnel involved in the national TB surveillance system in Malaysia for their dedication and give special thanks to the Big Data Task Force, UiTM Selangor Branch for their kind help with data cleaning processes.

## Author Contributions

**Conceptualization:** Zatil Zahidah Sharani, Nurhuda Ismail, Siti Munira Yasin.

**Data curation:** Zatil Zahidah Sharani, Asmah Razali.

**Formal analysis:** Zatil Zahidah Sharani, Nurhuda Ismail, Muhamad Rodi Isa.

**Investigation:** Zatil Zahidah Sharani, Nurhuda Ismail.

**Methodology:** Zatil Zahidah Sharani, Nurhuda Ismail, Siti Munira Yasin, Muhamad Rodi Isa.

**Supervision:** Nurhuda Ismail, Siti Munira Yasin, Muhamad Rodi Isa.

**Validation:** Nurhuda Ismail, Asmah Razali, Mas Ahmad Sherzkawee.

**Writing – original draft:** Zatil Zahidah Sharani, Nurhuda Ismail.

**Writing – review & editing:** Nurhuda Ismail, Siti Munira Yasin, Muhamad Rodi Isa, Ahmad Izuanuddin Ismail.

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
