## [Decision Letter · Decision Letter 0]

22 Nov 2022

PONE-D-22-25828T-BACCO SCORE: A predictive scoring tool for tuberculosis (TB) loss to follow-up among TB smokersPLOS ONE

Dear Dr. Ismail,

Thank you for submitting your manuscript to PLOS ONE. After careful consideration, we feel that it has merit but does not fully meet PLOS ONE’s publication criteria as it currently stands. Therefore, we invite you to submit a revised version of the manuscript that addresses the points raised during the review process.

We look forward to receiving your revised manuscript.

Kind regards,

Yuvaraj Krishnamoorthy

Academic Editor

PLOS ONE

Journal Requirements:

2. In your statement, please include the full name of the IRB or ethics committee who approved or waived your study, as well as whether or not you obtained informed written or verbal consent. If consent was waived for your study, please include this information in your statement as well.

Reviewers' comments:

Reviewer's Responses to Questions

**Comments to the Author**

1. Is the manuscript technically sound, and do the data support the conclusions?

Reviewer #1: Yes

2. Has the statistical analysis been performed appropriately and rigorously? 

Reviewer #1: Yes

3. Have the authors made all data underlying the findings in their manuscript fully available?

Reviewer #1: Yes

4. Is the manuscript presented in an intelligible fashion and written in standard English?

Reviewer #1: Yes

5. Review Comments to the Author

Reviewer #1: The manuscript is well conceived and written and needs some major revisions to be acceptable for publication.

Major Concerns:

1. There is no reporting of how well the model performed in the internal validation cohort.

2. The HOsmer-Lemeshow is not a reliable test for calibration, authors should present calibration plots as illustrated in the TRIPOD statement for reporting of risk prediction models.

3. Early in the paper it is not clearly and consistently articulated that a simple binary indicator of 'loss to follow-up (LTFUP)' is the outcome being modeled. It is confusing with all the information presented on different outcome states. is the outcome LTFUP relative to the occurrence of all other outcome types?

4. cross-sectional: the study uses fixed, baseline values of predictors to calculate the occurrence of LTFUP, which happen at some point in the future, so the repeated use of cross-sectional seems not quite accurate

5. On line 29 no statement about how much data is missing, whether it can be assumed to be missing at random, and whether any form of imputation was used. If not, how was this justified?

6. Process of multivariable model selection should be in the stats section rather than in results.

7. On line 188 test of equality for two AUC values should use DeLong's test rather than chi-square.

8. The tiny fraction of women in the study suggests these results should be limited to males only.

Minor comments:

1. multivariable instead of multivariate when modeling a single outcome.

2. AUC provides fair, not good, discrimination.

3. Line 134 confusing.

4. lines 187-8, chi-square not appropriate for comparing two AUC.

5. line 220: need to define RM2160 which presumably is transparent to Malaysians.

6. Line 244 - state 14 variables retained but only 13 listed.

7. line 260: no mention of model performance in validation cohort.

8. lines 282-3: no comprehensive discussion or treatment of missingness

9. lines 288-9: that chi-square test does suggest anything about missing values being completely at random.

10. Table 5: okay to present Hosmer-Lemeshow as supplement to calibration plots.

11. line 351: need to define IVDU

6. PLOS authors have the option to publish the peer review history of their article (what does this mean?). If published, this will include your full peer review and any attached files.

Reviewer #1: No

---

## [Author Response · Author response to Decision Letter 0]

11 Jan 2023

Major Concerns:

1 There is no reporting of how well the model performed in the internal validation cohort. 

Reports on the performance of the internal validation cohort is available in the results section under the performance of final model.

2 The HOsmer-Lemeshow is not a reliable test for calibration, authors should present calibration plots as illustrated in the TRIPOD statement for reporting of risk prediction models:

As per suggestion, calibration plots have been added in addition to the Hosmer-Lemeshow value for the model calibration reporting. The calibration curve for the final model and the internal validation model are available in Fig 3 and Fig 4.

3 Early in the paper it is not clearly and consistently articulated that a simple binary indicator of 'loss to follow-up (LTFUP)' is the outcome being modeled. It is confusing with all the information presented on different outcome states. is the outcome LTFUP relative to the occurrence of all other outcome types?

The outcome being measured and modelled in this study is the LTFU. The LTFU outcome is being compared with the successful TB outcome (cured and completed treatment)

4 Cross-sectional: the study uses fixed, baseline values of predictors to calculate the occurrence of LTFUP, which happen at some point in the future, so the repeated use of cross-sectional seems not quite accurate: 

We have look back at the methods of this study. We agree that the term “cross-sectional study” used in this study is inappropriate. This study involves secondary data analysis of a cohort of TB patients from year 2013-2017 in the national TB database. We looked back at the predictors that lead to the LTFU outcome. Thus, the correct study design for this study would be a retrospective cohort study.

5 On line 29 no statement about how much data is missing, whether it can be assumed to be missing at random, and whether any form of imputation was used. If not, how was this justified? 

Additional statement on the missing data information has been added in the abstract and result section. We have run the missing value analysis and the little MCAR test to report on the data missingness. Certain variable in the database has been recategorized and subsequent changes on the results and figure has been amended.

6 Process of multivariable model selection should be in the stats section rather than in results:

The paragraph on variable selection for the final model has been changed to the methodology section under the statistical analysis part.

7 On line 188 test of equality for two AUC values should use DeLong's test rather than chi-square:

As per suggestion we have analyse the performance between the two-model using the DeLong’s test. The outcome of this test is included under the result section in the performance of the final model paragraph.

8 The tiny fraction of women in the study suggests these results should be limited to males only:

The prevalence of woman who smokes in our country based on the latest national health morbidity survey (NHMS, 2015) is 1.3% which correspond to our finding in this study. Thus, we choose to remain the female subject in this study. Besides that, gender was not a significant predictor in this study, therefore the small proportion of female gender do not affect the performance of the model.

Minor comments:

1 Multivariable instead of multivariate when modeling a single outcome: The term multivariate used in this manuscript has been changed to multivariable throughout the manuscript.

2 AUC provides fair, not good, discrimination:

Correction has been made. The term used on the AUC has been changed to fair discrimination.

3 Line 134 confusing:

The sentence has been rearranged. The outcome being measured in this study is the LTFU TB outcome.

4 Lines 187-8, chi-square not appropriate for comparing two AUC:

As per suggestion we have analyse the performance between the two-model using the DeLong’s test. The outcome is available under the result section.

5 line 220: need to define RM2160 which presumably is transparent to Malaysians:

The categorization used for the income level is according to the median personal income for the population of Selangor state based on the department of statistic Malaysia (DOSM, 2018). Details on this variable has been added under the statistical analysis paragraph.

6 Line 244 - state 14 variables retained but only 13 listed:

We have corrected the sentence. The correct number of variables included in the multivariable analysis were 14, and 12 were retained in the final model.

7 line 260: no mention of model performance in validation cohort:

The performance of the Internal validation model was mentioned under the result section in the performance of the final model paragraph. 

8 lines 282-3: no comprehensive discussion or treatment of missingness Management and reports on the missing data has been included in the result section under the prognostic scoring system for LTFU paragraph.

9 lines 288-9: that chi-square test does suggest anything about missing values being completely at random:

We have conducted the missing value analysis and the little’s MCAR test to handle the missing data in the database. The report on the chi-square test has been corrected.

10 Table 5: okay to present Hosmer-Lemeshow as supplement to calibration plots:

Calibration curve for both final and internal validation model has been plotted under Fig 3A and Fig 3B in the manuscript using GGplot R statistical package.

11 line 351: need to define IVDU

IVDU has been defined as intravenous drug user in the sentence.

Thank you

---

## [Decision Letter · Decision Letter 1]

8 May 2023

PONE-D-22-25828R1T-BACCO SCORE: A predictive scoring tool for tuberculosis (TB) loss to follow-up among TB smokersPLOS ONE

Dear Dr. Ismail,

Thank you for submitting your manuscript to PLOS ONE. After careful consideration, we feel that it has merit but does not fully meet PLOS ONE’s publication criteria as it currently stands. Therefore, we invite you to submit a revised version of the manuscript that addresses the points raised during the review process.

We look forward to receiving your revised manuscript.

Kind regards,

Yuvaraj Krishnamoorthy

Academic Editor

PLOS ONE

Reviewers' comments:

Reviewer's Responses to Questions

**Comments to the Author**

1. If the authors have adequately addressed your comments raised in a previous round of review and you feel that this manuscript is now acceptable for publication, you may indicate that here to bypass the “Comments to the Author” section, enter your conflict of interest statement in the “Confidential to Editor” section, and submit your "Accept" recommendation.

Reviewer #1: All comments have been addressed

Reviewer #2: (No Response)

2. Is the manuscript technically sound, and do the data support the conclusions?

Reviewer #1: Yes

Reviewer #2: (No Response)

3. Has the statistical analysis been performed appropriately and rigorously? 

Reviewer #1: Yes

Reviewer #2: (No Response)

4. Have the authors made all data underlying the findings in their manuscript fully available?

Reviewer #1: Yes

Reviewer #2: (No Response)

5. Is the manuscript presented in an intelligible fashion and written in standard English?

Reviewer #1: Yes

Reviewer #2: (No Response)

6. Review Comments to the Author

Reviewer #1: The authors have done a nice job responding to all my previous concerns. I advise acceptance with minor revisions and do not need to see the manuscript again.

1. On line 51 put a comma in 835,000.

2. re-vise line 178 as follows: ' ... with LTFU were analyed using ... '

3. revise line 183 as ' For the monthly income level, the cutpoint at ... (488.30 USD) was based on ... '

4. Line 227: homogeneous

5. In Fig 2 lease add the labelled AUC for each curve.

6. For Fig 3 recommend confidence limits for blue model performance to show overlap with diagonal.

Reviewer #2: Authors intended to develop a prognostic scoring tool in predicting loss to follow-up (LTFU) among TB smoking patients. The constructed scoring was based on T-BACCO SCORE and predicted the risk for LTFU with low-, medium- and high-risk. The results showed a fair discrimination, good calibration.

1. The outcome is loss to follow-up. What’s the length of follow-up time for each individual? Are they very different or the same across the individuals? If vary, will this confound the results?

2. Line 179. The significance level was set to be 0.025. Please explain why this threshold is selected?

3. Line 194. AUC of 0.5 is selected as a discrimination threshold. However, 0.5 is just the value with random classification. Will such a low threshold be a concern?

4. Line 199. De Long test was used to compare the AUC between models. Please clarify what two models were compared? development and the internal validation cohort? If so, does it really make sense to compare their AUC when they are using different data?

5. It’s good to see the missingness doesn’t differ between development cohort and internal validation cohort. However, shouldn’t the missing data be taken care before split the sample into two cohorts?

7. PLOS authors have the option to publish the peer review history of their article (what does this mean?). If published, this will include your full peer review and any attached files.

Reviewer #1: No

Reviewer #2: No

---

## [Author Response · Author response to Decision Letter 1]

18 May 2023

Response to Reviewers

Dear PLOS One Editors,

AMENDMENT TO THE MANUSCRIPT ENTITLED “T-BACCO SCORE: A PREDICTIVE SCORING TOOL FOR TUBERCULOSIS(TB) LOSS TO FOLLOW-UP AMONG TB SMOKERS

Response to Reviewer 1:

1 On line 51 put a comma in 835,000

Response: Correction has been made accordingly on-line 51

2 Revise line 178 as follows: ' ... with LTFU were analysed using ... ' 

Response: The sentence has been corrected on line 178 “with LTFU were analysed using a simple logistic regression analysis”.

3 Revise line 183 as ' For the monthly income level, the cut point at ... (488.30 USD) was based on ... ' 

Response: The cut of point at RM2160 (488.30 USD) used was based on the median personal income of population in this country according to the department of statistics Malaysia in 2018 (DOSM).

4 Line 227: homogeneous 

Response: Spelling correction has been made accordingly, from “homogenous” to “homogeneous”.

5 In Fig 2 please add the labelled AUC for each curve.

Response: Amendment has been made in Fig 2. AUC value for each curve has been added.

6 For Fig 3 recommend confidence limits for blue model performance to show overlap with diagonal. 

Response: Calibration curve Fig 3(A) and Fig 3(B) has been amended. The confidence limits each calibration curve was inserted as per-recommendation.

Response to Reviewer 2:

1 The outcome is loss to follow-up. What’s the length of follow-up time for each individual? Are they very different or the same across the individuals? If vary, will this confound the results?

Response: The length of follow-up time unlikely confound the results. Majority of each individual follow-up duration is limited to the recommended six-month of TB treatment duration. Patients with TB MDR status (who needs longer treatment duration) has been excluded from the databased.

2 Line 179. The significance level was set to be 0.025. Please explain why this threshold is selected? 

Response: For variable selection into the multiple logistic regression analysis, significance level was set at 0.25 (not 0.025). This threshold was selected to ensure that adequate variable can be considered in the final logistic model. 

3 Line 194. AUC of 0.5 is selected as a discrimination threshold. However, 0.5 is just the value with random classification. Will such a low threshold be a concern?

Response: We are aware that AUC of 0.5 indicates no discrimination and AUC 0.7 is considered acceptable. However, in clinical setting within the TB domain especially on LTFU issue an AUC more than 0.6 is considered satisfactory as compared to the common acceptable threshold of 0.7. We have revised the sentence in line 194 to “AUC value ranges from 0.5 to 1, where a greater AUC value indicates that the model has better ability to distinguish patients who were LTFU and had successful TB treatment outcome”.

4 Line 199. De Long test was used to compare the AUC between models. Please clarify what two models were compared? development and the internal validation cohort? If so, does it really make sense to compare their AUC when they are using different data?

Response: Thank you for your comment. Yes, the two model that were compared are the development and the internal validation cohort. Initially, De Long test was used to inform that the performance between the two model has not much difference. However, after careful reading, we realized that De Long test is used to compare two correlated model. Therefor this analysis will be removed from this study finding. The comparison of the ROC between the two cohort will be assessed based on their overlapping confidence interval:

Development cohort: 0.681 (95% CI 0.657-0.713)

Internal validation cohort: 0.668 (95% CI 0.639-0.698)

5 It’s good to see the missingness doesn’t differ between development cohort and internal validation cohort. However, shouldn’t the missing data be taken care before split the sample into two cohorts?

Response: We agreed that missing data analysis should be managed earlier, however the two cohort (development and the internal validation) are coming from the same source of data based and it was divided randomly, therefor it should not be a concern. The purpose of analysing the missing data separately is to provide a more representative picture on the data missingness for each cohort to the reader.

---

## [Decision Letter · Decision Letter 2]

5 Jun 2023

T-BACCO SCORE: A predictive scoring tool for tuberculosis (TB) loss to follow-up among TB smokers

PONE-D-22-25828R2

Dear Dr. Ismail,

We’re pleased to inform you that your manuscript has been judged scientifically suitable for publication and will be formally accepted for publication once it meets all outstanding technical requirements.

Kind regards,

Yuvaraj Krishnamoorthy

Academic Editor

PLOS ONE

Additional Editor Comments (optional):

Reviewers' comments:

Reviewer's Responses to Questions

**Comments to the Author**

1. If the authors have adequately addressed your comments raised in a previous round of review and you feel that this manuscript is now acceptable for publication, you may indicate that here to bypass the “Comments to the Author” section, enter your conflict of interest statement in the “Confidential to Editor” section, and submit your "Accept" recommendation.

Reviewer #1: All comments have been addressed

Reviewer #2: All comments have been addressed

2. Is the manuscript technically sound, and do the data support the conclusions?

Reviewer #1: Yes

Reviewer #2: (No Response)

3. Has the statistical analysis been performed appropriately and rigorously? 

Reviewer #1: Yes

Reviewer #2: (No Response)

4. Have the authors made all data underlying the findings in their manuscript fully available?

Reviewer #1: No

Reviewer #2: (No Response)

5. Is the manuscript presented in an intelligible fashion and written in standard English?

Reviewer #1: Yes

Reviewer #2: (No Response)

6. Review Comments to the Author

Reviewer #1: (No Response)

Reviewer #2: (No Response)

7. PLOS authors have the option to publish the peer review history of their article (what does this mean?). If published, this will include your full peer review and any attached files.

Reviewer #1: No

Reviewer #2: No

---

## [Editor Report · Acceptance letter]

8 Jun 2023

PONE-D-22-25828R2 

T-BACCO SCORE: A predictive scoring tool for tuberculosis (TB) loss to follow-up among TB smokers 

Dear Dr. Ismail:

I'm pleased to inform you that your manuscript has been deemed suitable for publication in PLOS ONE. Congratulations! Your manuscript is now with our production department. 

Kind regards, 

on behalf of

Dr. Yuvaraj Krishnamoorthy 

Academic Editor

PLOS ONE